# Effect of Acute Exposure to the Ionic Liquid 1-Methyl-3-octylimidazolium Chloride on the Embryonic Development and Larval Thyroid System of Zebrafish

**DOI:** 10.3390/ani12111353

**Published:** 2022-05-25

**Authors:** Weikai Ding, Yangli Chen, Yousef Sultan, Junguo Ma, Yiyi Feng, Xiaoyu Li

**Affiliations:** 1Henan International Joint Laboratory of Aquatic Toxicology and Health Protection, College of Life Science, Henan Normal University, Xinxiang 453007, China; dweikai1029@163.com (W.D.); chen2787657141@163.com (Y.C.); sultanay18@hotmail.com (Y.S.); mjunguo_1378@126.com (J.M.); 2020040@htu.edu.cn (Y.F.); 2Department of Food Toxicology and Contaminants, National Research Centre, Dokki, Cairo 12622, Egypt

**Keywords:** [C8mim]Cl, zebrafish, thyroid, toxicity

## Abstract

**Simple Summary:**

In this study, we aimed to evaluate the effect of acute exposure to the ionic liquid 1-methyl-3-octylimidazolium chloride on the embryonic development and larval thyroid system of zebrafish. The results showed that the fish embryonic development, thyroid hormone level, and expression of HPTs-related genes were altered, suggesting that the ionic liquid [C8mim]Cl might pose an aquatic environmental threat to fish.

**Abstract:**

Previous studies have shown that ILs can induce toxicity in animals, plants, and cells. However, the effect of imidazolium-based ILs on the hypothalamus–pituitary–thyroid (HPT) axis of fish remains unknown. The present study aimed to evaluate the acute effect of [C8mim]Cl on the embryonic development and thyroid-controlled internal secretion system of zebrafish by determining the thyroid hormone level and the expression of HPT-related genes. The results obtained for embryonic developmental toxicity showed the survival rate, heart beats, and body length of fish had decreased 96 h after exposure to [C8mim]Cl, but the hatching rate had increased by the 48 h time point. The transcription levels of HTP-related genes showed that the genes *dio3*, *tg*, *ttr*, *tsh*, *trhrα*, *trhrβ*, *trhr2*, and *tpo* were up-regulated, while the expression levels of *dio1*, *trh*, *tshr*, and *nis* were significantly suppressed. Furthermore, we found that exposure to [C8mim]Cl induced an alteration in the levels of thyroid hormones that increased the T3 but decreased the T4 content. In conclusion, our study indicated that acute exposure to [C8mim]Cl altered the expression of HTP-related genes and disturbed the thyroid hormone level, suggesting that the ionic liquid [C8mim]Cl might pose an aquatic environmental threat to fish.

## 1. Introduction

Ionic liquids (ILs) are formed by an organic cation together with an organic or inorganic anion and are mostly defined as room-temperature molten salts, since they have melting points below 100 °C. Due to their unique characteristics, including low vapor pressure, heat resistance, and strong electrical conductivity, they have attracted the attention of investigators [1,2]. ILs are claimed to be nontoxic to the environment and therefore can be used as a replacement for traditional organic solvents in chemical catalysis, chemical analysis, agricultural production, and even pharmaceutical synthesis [3,4,5]. Unfortunately, in recent years, they have been found to pose a threat to the ecosystem because of uncompleted degradation by bacteria; subsequently, they cannot be considered “green” or “eco-friendly” [6]. Recently, different concentrations of ionic liquids were found in soils near urban landfill waste sites in Northumberland, UK; this was attributed to the lack of protective measures in the management or application of ILs [7]. Little information about the negative impact of ionic liquids in the environment, particularly in fish, is available; thus, this area requires further study.

An increasing number of studies have demonstrated that ILs have harmful effects on the cells of vertebrates both in vivo and in vitro. Yu et al. [8] found that the acute toxicity caused by l-octyl-3-methylimidazolium bromide ([C8mim]Br) exposure to mouse liver induced a significant up-regulation of the activity of antioxidant enzymes. Likewise, Li et al. reported that the acute exposure of brocaded carp (*Cyprinus carpio* L.) to 300 mg L^−1^ of [C8mim]Br for 7 days led to the suppression of their immune system and tissue damage [9]. The chronic exposure of silver carp to [C8mim]Br for two months markedly changed the oxidation and caused inflammation and hepatocellular apoptosis in the spleens of fish [10,11]. Although some previous studies have shown that subacute exposure to ILs could alter the weight of the pituitary glands of rats [12], at present, to the best of our knowledge, no work can be found regarding the potential effect of ILs on the thyroid-controlled internal secretion system of zebrafish.

The thyroid hormones (THs) play a key role in maintaining the homeostasis of the internal environment in vertebrates and regulating the metabolism, growth, development, and survival of individuals [13,14,15]. The synthesis of thyroid hormones is mainly achieved through the regulation of the hypothalamus–pituitary–thyroid (HPT) axis [16]. Meanwhile, it is thought that fish embryo development and larval–juvenile growth are both regulated by the HPT axis [17]. The hypothalamus, as the center of the neuroendocrine system, mainly takes part in the secretion of thyrotropin-releasing hormone (trh) and integrates neuromodulation with body fluid regulation through its connection to the pituitary gland. Trh can stimulate the anterior pituitary to secrete a thyroid-stimulating hormone (tsh), which is indispensable to the generation of thyroxine in the thyroid follicles. In vertebrates, triiodothyronine (T3) and tetraiodothyronine (T4) are two THs and they often rely on thyroid hormone receptors (trs) in TH-responsive tissues for their biological functions [18]. A number of previous studies have shown that exposure to environmental pollution could disrupt the levels of THs in humans and animals, resulting in growth retardation [19,20]. Even though there are increasing concerns regarding the ability of ILs to disrupt the endocrine system, no previous studies on the effects of these substances on TH production and adverse outcomes have been performed.

The present study aimed to evaluate the effect of the imidazolium-based ionic liquid 1-methyl-3-octylimidazolium chloride ([C8mim]Cl) on the thyroid-controlled internal secretion system of zebrafish by the determination of the thyroid hormone level and the expression of HPT-related genes.

## 2. Materials and Methods

### 2.1. Chemicals

[C8mim]Cl (CAS No. 64697-40-1, purity >99%) was obtained from Qingdao Aolike New Material Technology Co., Ltd., Qingdao, China. Stock solutions of the ionic liquid (1000 mg L^−1^) were prepared using ultra-pure water and stored at 4 °C. ELISA kits for thyroid hormones T3 and T4 were purchased from Shanghai Enzyme-linked Biotechnology Co., Ltd., Shanghai, China. All other chemicals used in the current study were of analytical grade.

### 2.2. Zebrafish and Experimental Design

Adult AB-wild type zebrafish (Danio rerio, 120 days old) were obtained from the China Zebrafish Resource Center (Wuhan, China), maintained in zebrafish housing systems, and fed the eggs of commercial brine shrimp twice a day. The following conditions were applied: circulating water temperature 28.0 ± 0.5 °C, water conductivity <30 μS/cm, water hydrogen ion concentration 7.0–7.4, water-dissolved oxygen 7.0 ± 0.2 mg L^−1^, sodium chloride 0.25‰, and ammonia nitrogen content <0.2 mg L^−1^. Fish were subjected to 14 h of light and 10 h of darkness routinely. On the eve of [C8mim]Cl exposure, zebrafish (two males and one female) were placed in a dedicated incubator with a barrier plate for separation. Twelve hours later, the barrier plate was removed for natural fertilization. Using stock solution, different concentrations of [C8mim]Cl were prepared (5.08, 10.16, and 20.32 mg L^−1^, representing 1/10 of LC50, 1/5 of LC50, and 2/5 of LC50 for embryo lethality at 96 h, respectively) in aerated tap water. In triplicate, 150 embryos were randomly distributed in each glass Petri dish (10 cm in diameter) containing 150 mL of exposed medium. The control group was exposed to aerated tap water. Those embryos were exposed for 96 h, and the medium was replaced with a new solution every 24 h to maintain the stability of the exposure concentration. The dead embryos were removed and counted daily. The hatching rate was recorded at 48 h and 72 h. The heartbeat rate for 30 s and the length of the larvae were also recorded using a stereoscopic microscope (Olympus SZ61, Tokyo, Japan) by an accessory software according to a previously described experimental method [21] 

### 2.3. Total RNA Extraction and Quantization RT-PCR

At the exposure point of 96 h, 30 larval zebrafish from each treatment were randomly selected for total RNA extraction according to the operation manual (TakaRa, Beijing, China). The concentration of RNA was detected at 260 and 280 nm using a Nanodrop One microvolume UV–Vis spectrophotometer (Thermo Scientific, Waltham, MA, USA). RNA with a ratio between 1.8 and 2.0 was applied in the following experiments; meanwhile, RNA integrity was measured using 1% agarose gel electrophoresis. The HiFiScript cDNA Synthesis Kit (Cowin Bio., Beijing, China) was used for the synthesis of the first chain of cDNA and genomic DNA removal following the previous study [22].

The special primer sequences and Genebank accession numbers of HPT axis regulatory genes are displayed in Table 1. The beta actin (β-actin) gene was used as a reference gene. The 20 μL PCR reaction system consisted of 2 μL of cDNA template, 0.4 μL of forward primer, 10 μM of reverse primer, 10 μL of MonAmp SYBR Green qPCR Mix (2×) (Monad Biotech Co., Ltd., Wuhan, China), and 7.2 μL of nuclease-free water. The reaction conditions were as follows: pre-heating at 95 °C for 5 min; degeneration at 95 °C for 10 s and anneal-elongation at 72 °C for 30 s, both repeated for 40 cycles; and elongation at 72 °C for 1 min. The expression of genes was quantified using the LightCycler^®^ 96 (Roche, Basel, Switzerland) real-time PCR system. All the experimental samples were repeated three times for analysis. The unique peak of the dissolved curve was used as a marker of specificity for each pair of primers. The relative mRNA expression level of HPT axis genes was calculated using the 2^−∆∆ct^ method [23].

### 2.4. Thyroid Hormone Determination

At 96 h, 40 larvae zebrafish from each experimental group were collected and measured to determine their content of thyroid hormone (T4 and T3) using commercial ELISA kits purchased from Shanghai Enzyme-linked Biotechnology Co., Ltd., Shanghai, China. The concentration standard curves and absorbance values were established according to the experimental instructions of the manufacturers. In brief, the whole larvae were homogenized on ice using PBS at a ratio of 1:9 with the aid of an electric tissue grinder (Tiangen Biotech Co., Ltd., Beijing, China). The samples were centrifuged at 3000 rpm for 20 min at 4 °C. Sample supernatant was used to measure the concentrations of T4 and T3. The minimum detection values were 1 ng mL^−1^ and 3 ng mL^−1^ for T3 and T4, respectively.

### 2.5. The Statistical Analysis

In this study, the data were calculated, analyzed, and visualized using Microsoft 365 Excel (Microsoft, Redmond, DC, USA), SPSS statistics 20.0 (IBM, Armonk, NY, USA), and GrapdPad Prism 8.0.2 (GraphPad Software, San Diego, CA, USA), respectively. The results were represented as mean ± the standard error of the mean (SEM). Significant differences between the treatments and the control group were calculated by one-way ANOVA [24]. In addition, Dunnett’s test was used in post hoc pairwise comparisons between the control group and the [C8mim]Cl exposure group. *p* values < 0.05 (“*”) and < 0.01 (“**”) were statistically significant.

## 3. Results

### 3.1. Effect of [C8mim]Cl Exposure on the Embryonic Development of Zebrafish

The effect of [C8mim]Cl exposure on zebrafish development is shown in Figure 1. Compared with the control group, exposure to the examined concentrations of [C8mim]Cl for 96 h significantly decreased the survival rate, heart rate, and body length of zebrafish (Figure 1A,C,D). The concentrations 10.16 and 20.32 mg L^−1^ of [C8mim]Cl significantly increased the hatching rate after 48 h, whereas no effect was observed after 72 h of exposure.

### 3.2. Effect of [C8mim]Cl on the Expression of HPT Axis-Related Genes

The mRNA expression levels of the HPT axis genes of larvae after exposure to different concentrations of [C8mim]Cl for 96 h are shown in Figure 2. The qPCR results indicated that the levels of expression of *dio1*, *nis*, and *trh* in the group treated with 20.32 mg L^−1^ were significantly lower than those of the control group. However, the levels of expression of *dio3*, *thra*, *tpo*, *tg*, *tsh* and *ttr* were remarkably promoted in the groups exposed to 10.16 and 20.32 mg L^−1^ of [C8mim]Cl. We also determined the expression of the thyrotropin-releasing hormone receptor family genes in zebrafish embryos and larvae; the three examined genes (*trhrα*, *trhrβ*, and *trhr2*) were significantly up-regulated in the groups treated with 10.16 and 20.32 mg L^−1^ when compared to the control group (Figure 3).

### 3.3. Thyroid Hormone Content in Larvae

The thyroid hormone contents from the whole bodies of zebrafish larvae after exposure to [C8mim]Cl for 96 h are illustrated in Figure 4. Exposure to 5.08 and 10.16 mg L^−1^ of [C8mim]Cl resulted in a significant increase in the T3 content (Figure 4A). However, the T4 content was statistically decreased in the treatment groups when compared with that of the control group (Figure 4B). Therefore, the ratio of T3 to T4 in the groups administered 5.08 and 10.16 mg L^−1^ of [C8mim]Cl became higher than that of the control (Figure 4C).

## 4. Discussion

Ionic liquids are a type of new organic solvents that are used in various fields, including chemistry, agriculture, and pharmacy [25,26]. Despite there being quite a few reports focusing on their negative effects and toxicity in organisms, little work has been carried out on the endocrine systems of vertebrates. In the present study, zebrafish were used as an experimental model to evaluate the effect of [C8mim]Cl exposure on the HPT axis of fish and its functional maintenance. Our results relating to embryonic toxicity demonstrated that exposure to [C8mim]Cl led to a decrease in the survival rates and body lengths of fish. However, the hatching rate was promoted at the early stage of embryonic development (24 h), which might be because the ionic liquid increased the embryonic chorion permeability. A feedback mechanism between the TH and HPT axis is critical to improving the somatic growth, development, and metabolism of zebrafish during larval development [27]. T4 and T3 are the main forms of TH in fish. Although T4 has few direct responsibilities, T3 can directly infiltrate cells and function as a biologically active hormone. In the present study, we found that [C8mim]Cl exposure altered the contents of T3 and T4 in whole fish larvae—i.e., T3 content increased while T4 decreased. Consequently, the ratio of T3 to T4 in these larvae was higher than that of larvae in the control group. Previous studies have indicated that exposure to environmental pollutants can change the physiological indicators and thyroid hormone levels of zebrafish, especially their endocrine disruptors [28,29,30]. Our result suggests that ionic liquid might function as an endocrine disruptor in fish.

In vertebrates, the thyroid endocrine system plays a key role in maintaining normal levels of thyroid hormones, which are regulated by the HPT axis. Briefly, the HPT axis is responsible for controlling the synthesis, release, and metabolism of thyroid hormones [31,32]. In the upstream, trh plays a vital role in provoking the synthesis and release of tsh in the pituitary gland, while tsh regulates the proliferation of thyroid cells and the synthesis and secretion of thyroid hormones. In the current study, the gene expression levels of *trh* and *tsh* in larvae after exposure to different concentrations of [C8mim]Cl remarkably increased. Nevertheless, the T4 content decreased, which might be due to the occurrence of a negative feedback mechanism between the pituitary gland and thyroid. Wang et al. demonstrated that *tsh* transcription increased while T4 content decreased after zebrafish were exposed to Tris (1,3-dichloro-2-propyl) phosphate (TDCPP) [33]. Likewise, Kim and Ji reported that humidifier disinfectants could alter thyroid hormone expression levels and disrupt the transcription levels of HPT axis genes in zebrafish, including up-regulated *trh* genes and down-regulated T4 genes [34]. 

It is well known that trs consists of thyroid hormone receptor alpha (trα) and thyroid hormone receptor beta (trβ), which play a wide range of roles in growth and development, tissue differentiation, and substance metabolism in combination with T3 [35,36]. Previous studies have shown that exposure to environmental pollutants can alter the transcription level of trs genes [37,38]. In our study, the expression of both *trα* and *trβ* was enhanced in zebrafish after they were exposed to [C8mim]Cl. This result was in agreement with the increase in the expression of trα and trβ found in Mongolian racerunners after their exposure to diflubenzuron [39]. In addition, previous studies have indicated that the expression of the trs gene was changed in zebrafish and other vertebrates after their exposure to chemicals [29,34].

TH synthesis requires the participation and regulation of the genes *nis*, *tpo*, and *tg* [40]. Nis is a glycoprotein in the thyroid plasma membrane that functions as the molecular basis of iodine uptake by thyroid cells and plays a vital role in central nervous system development [41,42]. Our results showed that exposure to ionic liquid decreased the whole-body T4 content and suppressed *nis* expression, implying that transcriptional down-regulation of the *nis* gene may be responsible for the decrease in T4 content. Such an observation was consistent with the work of Tang et al. [43], who found that exposure to bisphenol AF decreases TT4 and FT4 levels and down-regulates the gene expression of nis in zebrafish larvae. T4 and T3 synthesis relies on iodide and tg tyrosine residue coupled with a tpo catalytic reaction [44,45]. Tpo is a crucial enzyme that catalyzes the synthesis of T4 and is directly involved in the activation of iodine [46]. The results of our study showed that the expression of both *tg* and *tpo* increased in zebrafish larvae after their exposure to [C8mim]Cl for 96 h. This result may be due to a compensatory response to the reduced level of T4. Other studies demonstrated that decreasing the whole-body T4 level was concomitant with the increase in the transcription level of the *tg* and *tpo* genes in zebrafish after exposure to tris [47], triphenyltin (TPT) [48], fipronil (FIP) [49], and dichlorodiphenyltrichloroethane (p,p’-DDE) [50].

The protein ttr plays a vital role in transporting T4 to the target tissues [51,52]. In this study, *ttr* gene transcription was enhanced in zebrafish larvae exposed to [C8mim]Cl. Similarly, studies have shown that exposure to tetrabromobisphenol A (TBBPA) or polybrominated diphenyl ethers (BDE209) enhances *ttr* gene expression but decreases the T4 content in zebrafish [53,54]. A possible reason for this result might be because of the occurrence of a compensation mechanism as a response to the increase in *ttr* expression. 

In this study, three different forms of deiodinase genes were found and measured in zebrafish: *dio1*, *dio2*, and *dio3*. These play an important role in the regulation of the TH level of peripheral tissues. Bioactive T3 production requires dio2 to catalyze the deiodination of T4, whereas dio3 is responsible for catalyzing the deiodination of T4 and T3, producing the non-bioactive T3 and 3,3′-diiodo-L-thyronine (T2), respectively. Dio1 has a negligible role in blood TH homeostasis, but is responsible for iodine recovery and TH degradation [55]. In the present study, the expression levels of *dio2* and *dio3* increased, while that of *dio1* decreased in the exposed fish, which was in line with the study by Van der Geyten et al. [56], who found that both dio2 activity and *dio2* transcription were enhanced in fish, whereas hyperthyroidism decreased. The expressional decrease in *dio1* may prevent the degradation of T3, leading to an increase in T3 content. Meanwhile, dio3 plays an important role in the TH inactivation pathway, causing an increase in the content of T4 and consequently a decrease in the T3 content. Therefore, the exposure to [C8mim]Cl may alter the deiodinase gene transcription, causing abnormalities in THs.

Thyrotropin-releasing hormone receptors play vital roles in discerning trh. In the current study, the *trhrα*, *trhrβ*, and *trhr2* genes of the *trhr* family showed similar levels of expressional up-regulation in the treated larvae. This result is consistent with those obtained in a previous study by Liu et al., who found that exposure to PCB153 and p,p’-DDE increases the transcriptional level of *trhr* in rats [46]. Furthermore, in this study, we also noticed that the mRNA level of *trh* was up-regulated. This indicates the importance of trhr in effective combination with trh and the stimulation of tsh synthesis in the anterior pituitary. Meanwhile, the results of our present study show that exposure to low concentrations of [C8mim]Cl may alter the gene expression levels of HTPs more easily than exposure to high concentrations of [C8mim]Cl, which may be related to hormesis.

## 5. Conclusions

In conclusion, our study showed that the acute exposure of zebrafish larvae to the ionic liquid [C8mim]Cl altered the transcription levels of HTPs-related genes and disturbed their thyroid hormone levels. This study might be valuable for providing a further explanation of the toxicity mechanism of ionic liquids in fish and an evaluation of the negative effects of ionic liquids.

## Figures and Tables

**Figure 1 animals-12-01353-f001:**
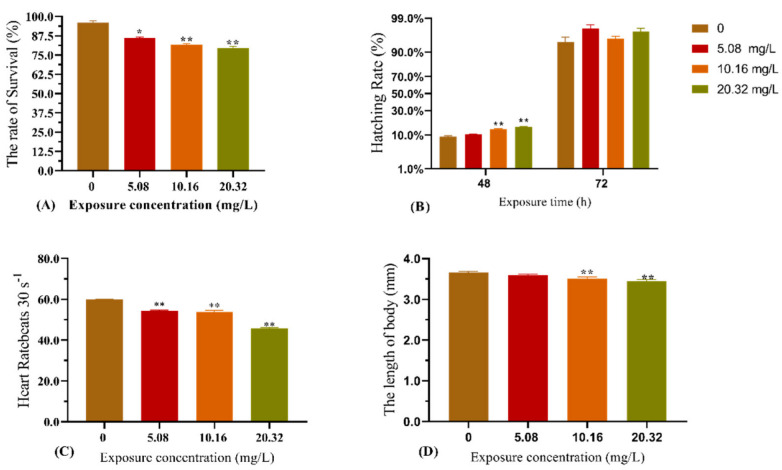
Effect of exposure to [C8mim]Cl (5.08, 10.16, and 20.32 mg L^−1^) for 96 h on the embryonic development of zebrafish. (**A**) The survival rate of zebrafish (96 h); (**B**) hatching rate of zebrafish after exposure for 48 h; (**C**) heart rate of zebrafish after exposure for 96 h; and (**D**) alteration of zebrafish body length after exposure (96 h). The results are represented as the means ± SEMs of three replicates. * *p* < 0.05, ** *p* < 0.01 expressed significant differences between the treatment group and the control group.

**Figure 2 animals-12-01353-f002:**
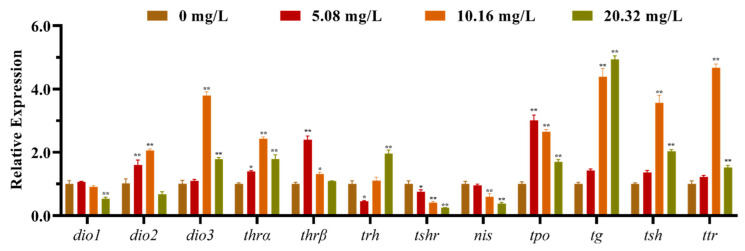
The mRNA expression levels of the hypothalamus–pituitary–thyroid axis genes in zebrafish embryos and larvae after exposure to 5.08, 10.16, and 20.32 mg L^−1^ of [C8mim]Cl for 96 h. The results are represented as the means ± SEMs of three replicates. * *p* < 0.05, ** *p* < 0.01 express significant differences between the treatment group and control group.

**Figure 3 animals-12-01353-f003:**
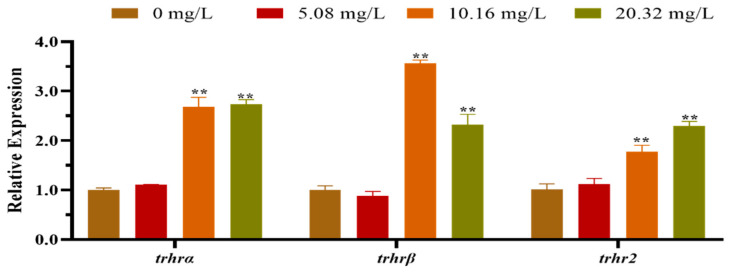
The mRNA expression levels of the thyrotrophin-releasing hormone receptor family genes in zebrafish embryos and larvae exposed to 5.08, 10.16, and 20.32 mg L^−1^ of [C8mim]Cl for 96 h. The results represented the means ± SEMs of three replicates. * *p* < 0.05, ** *p* < 0.01 express significant differences between the treatment group and control group.

**Figure 4 animals-12-01353-f004:**
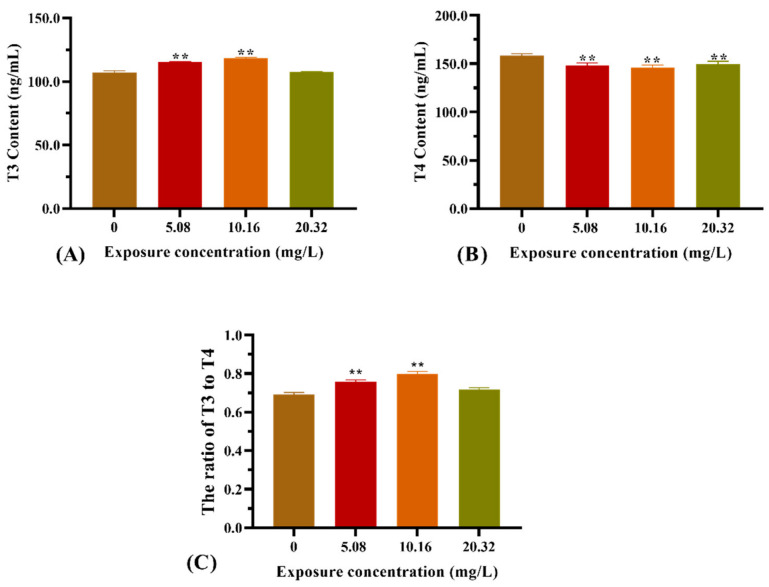
Effect of [C8mim]Cl exposure on the thyroid hormone levels of zebrafish larvae after 96 h. (**A**) T3 hormone, (**B**) T4 hormone, and (**C**) the ratio of T3 to T4. * *p* < 0.05, ** *p* < 0.01 express significant differences between the treatment group and control group.

**Table 1 animals-12-01353-t001:** The primer sequences of the hypothalamus–pituitary–thyroid axis genes tested in this study.

Gene	Sense Primers (5′-3′)	Accession No.
*β-actin*	FR 5′-TTCACCACCACAGCCGAAAGA-3′ RP 5′-TACCGCAAGATTCCATACCCA-3′	NM_131031.2
*dio1*	FR 5′-AGCGTGCACAAAAACTTGGAG-3′ RP 5′-TCCGATGCCTCCCTGATAGATA -3′	NM_001007283.2
*dio2*	FR 5′-GGCGGATGGTGGAGGAATT-3′ RP 5′-CACTGAGGAGGCAAGGAGAA-3′	NM_212789.4
*dio3*	FR 5′-AAGATGTTCACGCTGGAGTC-3′ RP 5′-GCTGCCGAAGTTGAGGATC-3′	NM_001177935.3
*nis*	FR 5′-TCTCTATGGCTTGCTGTTGGA-3′ RP 5′-GATGCTGTGGTGTTGAGTGTT-3′	XM_021478270.1
*tg*	FR 5′-CGTCCACCAATCGGAGACCTCA-3′ RP 5′-AACACCAGAACCGGCGCATTC-3′	NM_001329865.1
*thrα*	FR 5′-TATGGTGACGGAGGCTCATCGG-3′ RP 5′-ACTCGTGTGATGGCAGGCGTA-3′	NM_131396.1
*thrβ*	FR 5′- GCATCGCTGTTGGCATGGCTA-3′ RP 5′- ACTCTTCCTGTGTGGGCTCTGG-3′	XM_009306720.3
*tpo*	FR 5′-CTGCGGCGTGAAGGAGATA-3′ RP 5′-CCAATACAACGGCTTCCAACT-3′	XM_021467270.1
*trh*	FR 5′-GATACACCTGAGGAGTCCACTT-3′ RP 5′-CTGATCGTTCTCGCTGTTCTG-3′	NM_001012365.2
*trhr2*	FR 5′-GCATCCGCCAGATTCAGGTA-3′ RP 5′-TACAACCACCACAGCCAACA-3′	NM_001044346.1
*trhrα*	FR 5′-CGGAGACAGGTGACGAAGAT-3′ RP 5′-AGAGACGGCAGAACAGAAGG-3′	XM_682154.7
*trhrβ*	FR 5′-CACTCTTGCCACTGTCCTCTA-3′ RP 5′-TGTAACCTGTCTTCTGGATGCT-3′	NM_001114688.1
*tsh*	FR 5′- GCTGTCAACACCACCATCTG-3′ RP 5′-GTGAGGATCTGCATGTGAAGG-3′	NM_181494.2
*tshr*	FR 5′-ACGAGACAAGGCTAACAAGAGT-3′ RP 5′-TCAGGCTGTGGAAGGAATGC-3′	NM_001145763.2
*ttr*	FR 5′- CTCACTGTCCTCTGACGGTAA-3′ RP 5′- CACTTTCCCACTGGCAATCTT-3′	NM_001005598.2

## Data Availability

Data supporting the reported results are contained within the article.

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
