# Peer review of "Effect of Acute Exposure to the Ionic Liquid 1-Methyl-3-octylimidazolium Chloride on the Embryonic Development and Larval Thyroid System of Zebrafish"

_animals, 2022, doi:10.3390/ani12111353_

Round 1
Reviewer 1 Report
The study conducted by Weikai Ding et al. evaluated the acute effect 14 of 1-methyl-3-octylimidazolium chloride ([C8mim]Cl) on the thyroid-controlled internal secretion 15 system of zebrafish by determination of thyroid hormone level and the expression of the HPTs-16 related genes. Furthermore the authors evaluate the effect of [C8mim]Cl exposure on the fish 17 embryonic development. The results suggest that exposure of the ionic liquid [C8mim]Cl altered the transcription levels of HTPs-related genes and disturbed the thyroid hormone levels in zebrafish larvae. I think this is a very interesting study, however some questions should be considered in this study.
- In the abstract please write “IL” fully.
- There are work regarding the potential effect of ionic liquids on the thyroid-con- 59 trolled internal secretion system of other animal models? Please add.
- Please reduce introduction section on Th Hormone regulation.
- Why choose [C8mim]Cl? Please explain it and justify the choice of concentrations and exposure times.
- Please explain the reason why you chose zebrafish as your experimental model.
- The section “2.2 Zebrafish and experimental design” it's not entirely clear. Please clarify the experimental design.
- How was the survival rate, heart rate, and body length of zebrafish assesed? Please add hese information in Materials and Methods section.
- Increase the graphs resolution: the y-axis cannot be read.
- In this study are tested three different concentrations and two exposure times. Why was only the 96 hour time chosen for gene expression? In the discussion section, it is not clear which results are discussed, ie. related to which concentration? Considering that there is no direct dose-dependent relationship, which concentration seems to be more toxic and to regulate thyroid hormone regulation?
- Why does the higher concentration [C8mim]Cl give less gene regulation than the other two lower concentrations for which there seems to be a dose dependent effect? Please expalin in discussion section therse results.
Author Response
Please see the attachment. Thanks for your suggestion!

Reviewer 2 Report
The authors investigated the effect of acute exposure to the ionic liquid 1- methyl-3-octylim- idazolium chloride on the embryonic development and larval thyroid system of zebrafish. They designed four treatments to test the fish response to this toxicant. This manuscript (MS) was written and easy to understand. They covered a wide range of physical factors, from growth to gene expression. This work can increase our knowledge regarding fish response to this toxicant in the embryonic stage. However, some major issues significantly compromised the quality of this MS.
Major comments:
- First, the manuscript needs to be edited by a native English speaker to improve the language of the MS and fix errors. I corrected some of them, but still, more work needed to be done.
However, I have touched on some more points that can contribute to the improvement of this MS.
Minor comments
- Title, should be revised and represent the results and topic of this study better.
- Line 10, “alternate of traditional organic solvents” for which purpose?
- Line 12, please make sure you defined the abbreviations for the first time in the abstract as well as MS.
- Line 12-14, is confusing and there is no connection between points. Please revise it.
- Line 19 in which level?
- Line 23-24, again is not clear why this gene and why did you do this.
- Line 37-39, please revise it.
- Line 44, please avoid using this style “can’t” in scientific writing.
- Line 52, Ye et al. please add the year of this paper. Please read the papers to see how you should write this style of refs in the main text of MS.
- Line 88, please mention the novelty of your work in the last paragraph.
- Here and elsewhere, report P uppercase and italic (P<0.05).
- Throughout the MS, if there is no significant difference, no need to report P-value.
- Please reorder the keywords alphabetically and capitalize each word.
- Here and throughout the MS, please first mention the common name plus the scientific name, and for the rest of the MS, just report the common name.
- Please update the introduction with recent works as many studies are available from the last two years, which were not included in this section.
- Please review the literature much more carefully and cite more appropriate references.
Material and methods
- Well-organized section. Clear fellow and all required details were provided.
- Line 102, are you sure? Which prawn?
- Line 106-108, please revise it.
- Line 169-170, please provide more information regarding which kind of analysis you used.
- Please mention how many percentages of water were exchanged each day if you have monitored.
- For each analysis, please clarify how many fish were taken.
Results
- Well-written section, all necessary things have been covered.
- Line 173, revise and fix the error.
- Please provide background about the measured genes in this study in the Introduction section.
- For making more sense, please add the detected level on the environment in the introduction. Readers have no idea, for example, 5.08 is high or low.
- Please make the results more numeric by adding numbers in the parentheses.
- Line 223-27, please add more concepts here. It is not clear why you measured this.
Discussion
- Put the subheading for the discussion section like results. Also, keep a sequence in subheading for investigated factors, in material, and method, result, and discussion.
- Line 242-246, do you think increasing these hormones is negative or positive?. I suggest authors read more papers and get more knowledge regarding these hormones and then revise the discussion.
- Line 256, please check my comment regarding how you should write this kind of refs in the text of MS.
- Other parts of the discussion were written well and I have no comment.
- Some parts of the discussion are better updated with research in 2021 and 2022 as they refer to some old references. Please update the discussion with the latest studies as much as possible.
- Although you wrote this section well, you can still improve it by answering these questions and annotating them in the discussion section. Why were these results observed? Discuss more possible reasons.
- The conclusion needs to be revised and add more comprehensive concepts there. This section is too general and not impressive.
Best regards
Author Response
Please see the attachment. Thanks for your suggestions!

Round 2
Reviewer 1 Report
the authors responded adequately to comments by improving the manuscript.
Author Response
We have completed professional English language editing. Thanks for your suggestion.

Reviewer 2 Report
The authors improved the quality of the MS and I suggest authors reading one more time to fix few language errors. Then, it would be ready for the final steps for acceptance.
Author Response
We have completed professional English language editing. Thanks for your suggestion.

This manuscript is a resubmission of an earlier submission. The following is a list of the peer review reports and author responses from that submission.